# Association of Human Plasma Metabolomics with Delayed Dark Adaptation in Age-Related Macular Degeneration

**DOI:** 10.3390/metabo11030183

**Published:** 2021-03-21

**Authors:** Kevin M. Mendez, Janice Kim, Inês Laíns, Archana Nigalye, Raviv Katz, Shrinivas Pundik, Ivana K. Kim, Liming Liang, Demetrios G. Vavvas, John B. Miller, Joan W. Miller, Jessica A. Lasky-Su, Deeba Husain

**Affiliations:** 1Retina Service, Massachusetts Eye and Ear, Harvard Medical School, 243 Charles Street, Boston, MA 02114, USA; kevin.mendez@channing.harvard.edu (K.M.M.); Ines_Lains@meei.harvard.edu (I.L.); Archana_Nigalye@meei.harvard.edu (A.N.); Raviv_Katz@meei.harvard.edu (R.K.); Ivana_Kim@meei.harvard.edu (I.K.K.); Demetrios_Vavvas@meei.harvard.edu (D.G.V.); John_Miller@meei.harvard.edu (J.B.M.); Joan_Miller@meei.harvard.edu (J.W.M.); 2Channing Division of Network Medicine, Department of Medicine, Brigham and Women’s Hospital and Harvard Medical School, Boston, MA 02114, USA; jessica.su@channing.harvard.edu; 3Harvard Medical School, Boston, MA 02114, USA; janicekim93@gmail.com; 4Department of Ophthalmology, Schepens Eye Research Institute of Massachusetts Eye & Ear, Harvard Medical School, Boston, MA 02114, USA; Shrinivas_Pundlik@meei.harvard.edu; 5Program in Genetic Epidemiology and Statistical Genetics, Department of Epidemiology, Harvard T.H. Chan School of Public Health, Boston, MA 02114, USA; lliang@hsph.harvard.edu; 6Department of Biostatistics, Harvard T.H. Chan School of Public Health, Boston, MA 02114, USA

**Keywords:** age-related macular degeneration, dark adaptation, rod-intercept time, area under the dark adaption curve, metabolomics, mass spectrometry

## Abstract

The purpose of this study was to analyze the association between plasma metabolite levels and dark adaptation (DA) in age-related macular degeneration (AMD). This was a cross-sectional study including patients with AMD (early, intermediate, and late) and control subjects older than 50 years without any vitreoretinal disease. Fasting blood samples were collected and used for metabolomic profiling with ultra-performance liquid chromatography–mass spectrometry (LC-MS). Patients were also tested with the AdaptDx (MacuLogix, Middletown, PA, USA) DA extended protocol (20 min). Two measures of dark adaptation were calculated and used: rod-intercept time (RIT) and area under the dark adaptation curve (AUDAC). Associations between dark adaption and metabolite levels were tested using multilevel mixed-effects linear modelling, adjusting for age, gender, body mass index (BMI), smoking, race, AMD stage, and Age-Related Eye Disease Study (AREDS) formulation supplementation. We included a total of 71 subjects: 53 with AMD (13 early AMD, 31 intermediate AMD, and 9 late AMD) and 18 controls. Our results revealed that fatty acid-related lipids and amino acids related to glutamate and leucine, isoleucine and valine metabolism were associated with RIT (*p* < 0.01). Similar results were found when AUDAC was used as the outcome. Fatty acid-related lipids and amino acids are associated with DA, thus suggesting that oxidative stress and mitochondrial dysfunction likely play a role in AMD and visual impairment in this condition.

## 1. Introduction

Age-related macular degeneration (AMD) is a leading cause of blindness among people 50 years of age or older [1]. Visual acuity (VA) currently remains the gold-standard outcome in AMD; however, patients usually do not present with vision loss until late into the disease [2]. Other functional outcome measures have been proposed, such as contrast sensitivity, low-luminance visual acuity, photopic or scotopic light sensitivity, and dark adaptation (DA) [2,3]. DA, in particular, is promising as even in the setting of normal visual acuity, patients with early stages of the disease often report loss of night vision and reduced ability to adapt from brightly lit to dark environments [4]. Studies have shown that with the use of a commercially available device—the AdaptDx (MacuLogix, Harrisburg, PA, USA) dark adaptometer—dark adaptation can differentiate between patients with and without AMD, and across severity stages of the disease [2,3,5,6].

Since impaired DA has been shown to occur early on in AMD disease progression [7,8], it is likely that understanding the pathophysiology behind these changes can contribute to a better understanding of AMD pathogenesis and address the great unmet need for potential treatments for AMD. Our group has shown that metabolomics, the qualitative and quantitative analysis of metabolites, can provide important insights to address these needs [9,10,11]. The metabolome, defined as the set of all measurable small molecules in a biospecimen, is most closely related to phenotype and can provide information on the interactions between genetics, environment, and lifestyle that contribute to multifactorial diseases such as AMD. While there have been metabolomic-based studies that use AMD severity stage as the outcome [9,10,11], to our knowledge, no studies have considered associations with DA.

In this study, we analyzed the association between plasma metabolite levels and measures of dark adaptation in patients with AMD and controls, using rod-intercept time (RIT) and area under the dark adapation curve (AUDAC) as the measures of dark adaption. Our goal was to identify metabolites associated with dark adaptation in AMD to contribute to the current understanding of the mechanisms behind DA changes in AMD, and to elucidate the mechanisms involved in AMD pathophysiology.

## 2. Results

We included a total of 71 subjects: 18 controls and 53 with AMD (13 early AMD, 31 intermediate AMD, and 9 late AMD). Of the nine late AMD patients, five had geographic atrophy (GA) and four had choroidal neovascularization (CNV) with active lesions. Clinical and demographic characteristics of the study cohort are presented in Table 1.

For all analyses, we report *p*-values at two thresholds: *p*-values < 0.01 to denote a trend towards significance, and *p*-values < 0.0019 to denote statistically significant findings after accounting for multiple testing. The statistically significant threshold (*p*-value < 0.0019) was calculated based on the effective number of independent tests accounting for 80% variance (ENT80) [12,13]. Further details are provided in Section 4.7.

### Dark Adaptation and Plasma Metabolite Levels

Eight plasma metabolites were associated with RIT at a *p*-value of less than 0.01, including two amino acids (N-acetylglutamine and N-acetylleucine), one carbohydrate (mannitol/sorbitol), and five fatty acid-related lipids (Table 2). The fatty acid linoleate was statistically significant based on ENT80 (*p*-value < 0.0019). When AUDAC was considered as the outcome, as shown in Table 3, 14 plasma metabolites with *p*-values < 0.01 were seen, including the two amino acids and one carbohydrate significantly associated with RIT, as well as two nucleotides (beta-alanine and xanthine), and nine lipids (four sphingomyelins, two fatty acid-related lipids, one phosphatidylcholine, one lysophospholipid, and one hexosylceramide). Five of these metabolites had *p*-values below the ENT80 threshold, one carbohydrate (mannitol/sorbitol), three lipids (linoleoylcholine, glycosyl ceramide, 1-linoleoyl-2-linolenoyl-GPC), and one nucleotide (beta-alanine).

In the stratified analysis with patients who failed to reach RIT, we found three unique metabolites associated with AUDAC (*p*-value < 0.01), two amino acids related to tryptophan metabolism (indole-3-carboxylic acid and kynurenate), and one hexosylceramide lipid (glycosyl-N-palmitoyl-sphingosine (d18:1/16:0)) (Table 4). In the second stratified analysis with intermediate AMD patients, we found 11 metabolites with *p*-values less than 0.01, including five amino acids in the leucine, isoleucine, and valine metabolism sub-pathway, mannitol/sorbitol, one nucleotide (urate), one peptide (gamma-glutamylisoleucine), and three lipids (two fatty acid-related lipids and one lactosylceramide) (Table 5). Of these, three metabolites in the leucine, isoleucine, and valine metabolism pathway (3-methyl-2-oxovalerate, 3-methylglutaconate, and isoleucine) were statistically significant at a *p*-value threshold less than 0.0019 (ENT80).

## 3. Discussion

We present an analysis of the association between plasma metabolite levels and DA impairment in AMD patients and controls. After controlling for age, gender, body mass index (BMI), smoking status, race, Age-Related Eye Disease Study (AREDS) formulation supplementation, and AMD stage, we identified plasma metabolites associated with DA impairment. Key findings include the associations between dark adaptation with fatty acid-related lipids and the amino acids related to glutamate (N-acetylglutamine) and leucine, isoleucine and valine metabolism (N-acetylleucine) with both RIT and AUDAC as the outcome. Similar findings with fatty acid-related lipids were found in the stratified analysis with intermediate AMD patients, and while N-acetylleucine was not significant at a *p*-value less than 0.01, five other metabolites in the leucine, isoleucine and valine metabolism pathway were, suggesting these pathways play an important role driving dark adaptation impairment at this stage of disease.

The association between a decrease in various fatty acid conjugate bases or derivatives of polyunsaturated fatty acids (PUFAs) and impaired DA in AMD is particularly relevant as many studies have also found an association between lipid metabolism and AMD [9,11,14,15,16,17]. PUFAs of 20 carbons or more in length are mainly derived from linoleic and linolenic acid [17,18,19,20]. PUFAs are abundant in the retina and play important structural and functional roles [17,21,22,23], such as the differentiation and survival of photoreceptors, and the facilitation of visual perception [21,24]. Certain PUFAs have been correlated with a decreased risk of developing AMD [25,26,27], and others have shown promise as potential biomarkers for high-risk AMD patients [28,29]. In the process of dark adaptation, rhodopsin—a visual pigment consisting of opsin and 11-cis-retinal—photoisomerizes upon light absorption in the photoreceptor and converts 11-cis-retinal to an all-trans retinoid. The all-trans retinoid is released from opsin and conveyed to the retinal pigment epithelium (RPE), where it is re-converted to 11-cis retinal that recombines with opsin to regenerate rhodopsin [30]. While their exact role in dark adaptation is unknown, studies have found that certain PUFAs, such as docosahexaenoic acid (DHA), may be involved in rhodopsin regeneration due to their effect on interphotoreceptor retinal binding proteins (IRBP) transporting 11-cis-retinal from the RPE [31,32]. This suggests that oxidative stress in the retina—which is a possible mechanism for AMD [33]—correlates with a decrease in PUFAs through oxidation [33,34]. The subsequent decrease in PUFAs would then impair dark adaptation due to the inability of photoreceptors to regenerate rhodopsin as quickly [30,35]. This theory may provide further background on studies that found diets rich in PUFAs were associated with a lower risk of AMD [36,37,38,39,40]. Thus, we theorize that the levels of linoleic and linolenate (alpha or gamma)—precursors of longer-chained PUFAs—likely correlate with the concentration of PUFAs, which may be part of the pathogenic process of impaired DA in AMD.

The impact of oxidative stress on PUFAs may also explain why dark adaptation impairment occurs earlier than visual acuity impairment in AMD patients. Prior evidence suggests that the foveal regions tend to have lower concentrations of PUFAs, perhaps as a natural adaptation to decreased lipid peroxidation in regions with highest intensity light [31,41]. In AMD, the fovea is often spared until late stages of disease, with visual acuity impairment occurring late into the disease.

While less is known about the association between amino acids and AMD, metabolomic-based studies have found higher levels of various amino acids in AMD cohorts compared to controls [42,43,44,45]. Additionally, our group has previously found increased levels of amino acids including N-acetylasparagine, hypotaurine, beta-citrylglutamate, and N-acetylleucine in patients with AMD [9]. This study takes the association further by demonstrating increased levels of two amino acids, N-acetylglutamine and N-acetylleucine, specifically with impaired DA. While the role of these two amino acids in DA remains unclear, glutamine and leucine play important roles in various cellular processes. With regard to glutamine, other metabolomics studies have found elevated glutamine levels in AMD patients [10,14]. Glutamine is an amino acid necessary for immune competence and plays an important role through its antioxidant and cytoprotective effects [46]. Studies have shown that during high oxidative stress, glutamine is released by the skeletal muscle [47]. Again, as similarly noted by Kersten, it is not yet clear whether increased glutamine levels reflect increased immune system activation or decreased clearance of glutamine, among other possible mechanisms [14]. Leucine is a branched chain amino acid (BCAA) that promotes energy metabolism through fatty acid oxidation and mitochondrial biogenesis [48], among others [49]. Studies have shown that when mitochondrial function is impaired, concentrations of BCAAs in plasma can become markedly elevated [50,51,52]. Mitochondrial dysfunction has been proposed as a possible mechanism in macular degeneration, resulting in increased oxidative stress [53]. Altogether, this may further suggest that oxidative stress may play a role in the impairment of dark adaptation in AMD patients, although it remains to be investigated whether the increase in glutamine and leucine in the serum is a response to higher demand from the immune system or in response to other mechanisms.

Given the 20 min ceiling value for RIT, we performed an additional analysis using AUDAC as the measure of dark adaption. Similar to the initial findings, N-acetylglutamine and N-acetylleucine and fatty-acid related metabolites were significant. Additionally, there were four sphingolipids that were negatively associated with AUDAC. Given the role of sphingolipids in signaling pathways, multiple studies have suggested sphingolipids have a crucial role in the progression of AMD through the mediation of proliferation, survival, migration, neovascularization, inflammation, and death in retina cells [54,55,56,57]. As these sphingolipids were associated with AUDAC, and not RIT, it may suggest AUDAC may be a more sensitive functional measurement than RIT in measuring AMD progression due to its ability to measure DA beyond the 20 min ceiling for RIT. We then performed a stratified analysis looking at AUDAC for patients unable to reach RIT within the 20 min of testing, indicating those with worse retinal function. Of the three significant metabolites, two were amino acids in the tryptophan pathway. Tryptophan metabolism, as it relates to the kynurenine pathway, plays an important role in the synthesis of nicotinamide adenine dinucleotide (NAD^+^). As NAD^+^ is linked to cellular energy metabolisms, it is broadly related to inflammation [58]. Tryptophan metabolites were not found in our previous analyses, which may suggest that in the later stages of AMD progression, the pathophysiology related to severely impaired DA may be distinct.

In addition to these prior analyses, we specifically looked at the intermediate AMD cohort as it had the most individuals with a wide range of RIT values and could provide important insights into this stage of AMD. Similar to the overall findings, we found decreased levels of lipids and increased levels of amino acids with significant associations with RIT. Two of the three lipids were fatty acid-related (octadecanedioate (C18-DC) and linolenate (alpha or gamma; [18:3n3 or 6])), with the latter in common with the overall findings. All five amino acids, including 3-methylglutarylcarnitine, are part of leucine, isoleucine and valine metabolism. Furthermore, 3-methylglutarylcarnitine is known primarily for its diagnostic ability to detect a Reye’s-like syndrome, which is caused by the deficiency of Hydroxy-3-methylglutaryl-CoA lyase and prevents the body from processing leucine. Reye’s syndrome and Reye’s-like syndromes are known for generalized disturbances in mitochondrial metabolism and impaired fatty acid oxidation [59]. With studies showing that HMG-CoA lyase deficiency results in the disruption of redox homeostasis, which induces lipid peroxidation, oxidative damage, and mitochondrial dysfunction [60,61,62,63], this further suggests that oxidative stress and mitochondrial dysfunction are not only involved in AMD pathogenesis, but may be driving dark adaptation impairment. Additionally, specific to the intermediate stage AMD cohort, we found increased levels of nucleotides and peptides related to purine metabolism (urate, N6-succinyladenosine) in association with RIT. The association between purine metabolism and AMD has been previously demonstrated [11], and while there is growing evidence for the role in the retina regarding ATP release [64], the potential mechanisms associated with DA remain unclear.

The current study has a number of limitations, in particular, a relatively small sample size; however, this is the first time the association of metabolomics with DA in AMD has been demonstrated. Given the role of DA in AMD disease and its severity, the association findings that we report are important for furthering our understanding of AMD. Additionally, our AMD cohort was comprised primarily of white participants, with the intermediate stage AMD cohort comprised of solely white participants. This is in part related to the epidemiologic factors of AMD [65], and to the population that is generally served by the enrolling site of our tertiary care hospital. While we controlled for AREDS formulation supplement use due to direct effects of nutrition (i.e., supplementation) on the metabolome, these data were collected through self-reported questionnaires, opening them up to a potential response bias. Another limitation was the inability to perform suitable stratified analyses for AREDS formulation supplementation. This is due to the imbalance of AREDS formula supplement use by AMD stage (i.e., low supplementation in controls and early AMD, and high supplementation in intermediate and late AMD), combined with the small sample size (Table 1). Similarly, due to imbalance by AMD stage combined with small sample sizes, we were unable to perform suitable stratified analyses by reticular pseudodrusen, which has been previously reported to be associated with DA impairment [66]. This cross-sectional study was only a snapshot of patients’ metabolomes, which is highly dynamic and susceptible to external factors. Our study, although designed prospectively with standardized data collection, will require further longitudinal studies to confirm findings and assess how the metabolome changes in relation to visual impairment for AMD patients. Finally, with this relatively low sample size, we reported *p*-values < 0.01 to denote a trend towards significance, which increases the risk of false positive results. Given this limitation, we also provided results based on the ENT80 significance thresholds. Note, at an ENT50 or ENT80 cut-off, while the number of significant metabolites is reduced, the findings regarding fatty acid-related lipids and amino acids related to glutamate and leucine, isoleucine and valine metabolism remain consistent.

## 4. Materials and Methods

### 4.1. Study Design

This study is derived from a cross-sectional, prospective project on AMD biomarkers performed at the Department of Ophthalmology of Massachusetts Eye and Ear (MEE), Harvard Medical School, Boston, United States. It was conducted in accordance with the Health Insurance Portability and Accountability Act requirements and the tenets of the Declaration of Helsinki. The institutional review board of MEE approved this study and written informed consent was obtained from all participants.

### 4.2. Inclusion and Exclusion Criteria

From January 2015 to June 2016, we recruited and consented patients with an AMD diagnosis at their regular appointments [9]. Subjects were excluded if presenting with any other vitreoretinal diseases, active uveitis or ocular infection, significant media opacities that precluded observation of the ocular fundus, refractive error equal to or greater than 6 diopters of spherical equivalent, a formal diagnosis of glaucoma with a cup-to-disc ratio greater than 0.8, history of retinal surgery, history of any ocular surgery or intraocular procedure (such as laser or intraocular injections) within the 90 days before enrollment, or diagnosis of diabetes mellitus, with or without concomitant diabetic retinopathy. The control group consisted of subjects aged 50 years or older, without evidence of AMD in both eyes. The same exclusion criteria were applied.

### 4.3. Study Protocol

Study participants underwent a comprehensive eye examination. A standardized questionnaire was applied to all subjects, which included questions on demographics, past medical history, and current medication [67]. Additionally, for all participants, blood samples were collected into a sodium-heparin tube, which was centrifuged within 30 min (1500 rpm, 10 min, 20 °C) to obtain plasma for metabolomic analysis. Overnight fasting was required, and samples were collected within a maximum of 1 month after study inclusion.

### 4.4. AMD Grading

All AMD participants were imaged with non-stereoscopic, 7-field, color fundus photographs (CFP) (Topcon TRC-50DX; Topcon corporation, Tokyo, Japan) for diagnosis and grading. AMD was graded using the AREDS 2 study grading scheme [68,69]. Two independent graders, masked to all clinical data, analyzed field 2 CFP from all study participants for grading [68]. If there was a disagreement, a senior clinician (DH) established the final categorization. Before grading, images were standardized using software developed by our group [2], and then were evaluated with IMAGEnet 2000 software (version 2.56; Topcon Medical Systems, Oakland, NJ, USA). According to the most recent AREDS2 definitions [68], we defined the standard disc diameter as 1800 mm, which affects the size of the Early Treatment Diabetic Retinopathy Study (ETRDS) grid and of the standard drusen circles. Additionally, we considered that GA was present if a lesion had a diameter equal to or greater than 433 μm (AREDS circle I-2), and at least 2 of the following features were present: absence of retinal pigment epithelium (RPE) pigment, circular shape, or sharp margins (involvement of the central fovea was not required) [68].

Eyes were graded using the AREDS 2 scheme into 4 groups, which were used for statistical analysis [68]. These included: Control group (AREDS level 1)—presence of drusen maximum size < circle C0 and total area <C1; Early AMD (AREDS level 2)—rusen maximum size ≥ C0 but <C1 or presence of AMD characteristic pigment abnormalities in the inner or central subfields; Intermediate AMD (AREDS level 3)—presence of drusen maximum size ≥ C1 or drusen maximum size ≥ C0 if the total area occupied is >I2 for soft indistinct drusen and >O2 for soft distinct drusen; Late AMD (AREDS level 4)—presence of GA according to the criteria described above or evidence of neovascular AMD.

### 4.5. Dark Adaptation Testing

As described previously by our group [2,67], to avoid prior light exposure (from clinical examination and retinal imaging), DA was performed on a separate day, within a maximum time limit of 1 month after enrolling in the study. According to our protocol, the current refraction was confirmed for all study participants and it was optimized when needed. Patients were dilated to ≥6 mm, and DA was performed using the AdaptDx^®^ dark adaptometer (MacuLogix, Harrisburg, PA). Corrective lenses were introduced to account for the 30-cm viewing distance. Participants were given a 2-min demonstration test before the actual testing to familiarize them with the procedure. In the transition time from demonstration to test, the room lights were kept off.

DA testing was performed in the dark. Both eyes were tested separately, with the right eye tested first, and at least a 15-min rest period between eyes. During testing, the fellow eye was occluded with an eye patch. The extended protocol (20 min) was followed. First, eyes were bleached by exposure to a 505-nm flash for 0.8-ms at an intensity of 1.8 × 10^4^ scot cd/m^2^, which is equivalent to 76% bleaching level for rods. The flash of light passed through a square aperture sized to bleach a 6° area of the retina centered at 5° on the inferior visual meridian. Sensitivity measurements were started right after bleaching. The participant focused on the fixation light and would push a hand-held button when the stimulus light was visible. This stimulus light was a 505-nm, 2° circular test spot, located at 5° on the inferior visual meridian, which is anatomically 5° superior to the central fovea.

Sensitivity was then estimated using a 3-down, 1-up modified staircase threshold estimate procedure. The initial stimulus intensity was 5 scot cd/m^2^, which is the maximum stimulus intensity. The stimulus was presented every 2 or 3 s for a 200-ms duration. If the stimulus was detected, the patient was given 2 s to respond by pushing a response button. If the patient indicated that the stimulus was visible, the intensity was decreased for each successive presentation in steps of 0.3 log units until the subject stopped indicating that the stimulus was visible. Alternatively, if the patient indicated that the stimulus light was not visible, the intensity of the target was increased for each successive presentation in 0.1-log-unit steps until the patient responded that the stimulus light was once again visible. The intensity at which the light was again visible was defined as threshold. Successive threshold measurements started with the stimulus intensity of 0.2 log units brighter than the previous threshold measurement. The subject had a 15-s rest period between threshold measurements. If a threshold had a large deviation from prior thresholds, the measurement was considered unreliable, a fixation error was noted, and immediately an additional threshold was measured. Threshold measurements were made approximately once a minute for the duration of the DA test. The test would end when the patient’s sensitivity was twice consecutively measured to be greater than 5 × 10^−3^ scot cd/m^2^ or the test duration reached 20 min, whichever endpoint came first. The AdaptDx machine estimates the slope of the second component of rod-mediated dark adaptation and extrapolates the amount of time required to achieve a sensitivity of 5 × 10^−3^ scot cd/m^2^. This value is named RIT.

Due to the 20 min ceiling value for RIT, an additional measure, AUDAC, was included as an alternative measure of dark adapation in this analysis. In a standard representation of a dark adapation curve, x is the time (0 to 20 min) and y is the log sensitivity (0 to 3). To calculate AUDAC, we measured the area under the curve from the start of the test (time = 0, log sensitivity = 0) to when the log sensitivity of 3 was met, using a standard trapezoidal method [70]. A higher AUDAC indicates a higher delay in DA, elevated sensitivity threshold, or both.

### 4.6. Metabolomic Profiling and Data Processing

Plasma samples were stored in sterile cryovials at −80 °C and shipped to Metabolon, Inc^®^ in dry ice when all had been collected. Metabolomics profiling was performed using ultrahigh-performance liquid chromatography tandem mass spectrometry (MS) by Metabolon, Inc^®^ (617 Davis Drive, Suite 100, Morrisville, NC, USA) based on previously published protocols [9]. The data were then run through our standard quality control and data processing pipeline [9,71], where metabolite peak areas were then log-transformed and Pareto-scaled, with missing values imputed using a half-minimum approach. A total of 544 plasma metabolites were included in this analysis: 148 amino acids, 18 carbohydrates, 19 cofactors and vitamins, 8 energy metabolites, 308 lipids, 28 nucleotides, and 15 peptides.

### 4.7. Statistical Analysis

To analyze the association between plasma metabolite levels and dark adaptation, we used multilevel mixed-effects linear model through the R package “lme4” [72], accounting for the inclusion of two dark adaptation measures (i.e., left eye and right eye) and a single measure of plasma metabolite levels for each patient [2,4,9]. In these models, the outcome was RIT. For subjects failing to reach RIT within 20 min of testing, we assigned a value of 20 [2,4,66]. As a significant percentage of our patients had reached their ceiling value for RIT, we performed an additional analysis using AUDAC as the outcome. We then performed two stratified analyses. First, a stratified analysis for subjects who failed to reach RIT within the 20 min of testing using AUDAC as the outcome. Second, a stratified analysis for only patients with intermediate stage AMD given the importance of this stage of disease, combined with the wide range of RIT values and largest sample size (Table 1). Models were adjusted for age, smoking status, race, BMI, AMD stage, and AREDS formulation supplementation. Note, in both stratified analyses, all subjects were white.

For all analyses, we report the association *p*-values < 0.01 to denote a trend towards significance. In all result tables, we also report statistically significant findings based on the ENT to account for multiple testing [12,13]. We computed ENT80 as the number of principal components that were needed to explain 80% of the variance in the data. For ENT80, the significance threshold was 0.0019 (0.05/26).

## 5. Conclusions

In this study, we found that increased levels of leucine, isoleucine and valine metabolites and decreased levels of fatty acid-related lipids were associated with impaired DA in AMD. Similarly, metabolites in the same pathway were also associated with impaired DA when looking at the intermediate AMD cohort only. These findings suggest that oxidative stress and mitochondrial dysfunction may play an important role in driving AMD and visual impairment.

## Figures and Tables

**Table 1 metabolites-11-00183-t001:** Clinical and demographic characteristics of the study cohort.

Demographics and Clinical Characteristics	Control	Early AMD	Intermediate AMD	Late AMD
Number, n (%)	18 (25.4)	13 (18.3)	31 (43.7)	9 (12.7)
Number of eyes, n (%)	31 (24.8)	23 (18.4)	56 (44.8)	15 (12.0)
Included eye, n (%)				
OD	16 (51.6)	12 (52.2)	26 (46.4)	8 (53.3)
OS	15 (48.4)	11 (47.8)	30 (53.6)	7 (46.7)
Age, mean ± SD	65.7 ± 7.8	66.1 ± 9.3	70.4 ± 5.4	71.4 ± 6.9
Gender, n (%)				
Male	10 (55.6)	5 (38.5)	12 (38.7)	2 (22.2)
Female	8 (44.4)	8 (61.5)	19 (61.3)	7 (77.8)
BMI, mean ± SD	26.2 ± 3.8	26.7 ± 4.5	28.0 ± 4.3	29.6 ± 5.3
Race/ethnicity, n (%)				
White	16 (88.9)	10 (76.9)	31 (100.0)	7 (77.8)
Hispanic	1 (5.6)	2 (15.4)	0 (0.0)	2 (22.2)
Black	1 (5.6)	0 (0.0)	0 (0.0)	0 (0.0)
Asian	0 (0.0)	1 (7.7)	0 (0.0)	0 (0.0)
Smoking, n (%)				
Non-smoker	8 (44.4)	9 (69.2)	13 (41.9)	5 (55.6)
Ex-smoker	9 (50.0)	4 (30.8)	17 (54.8)	4 (44.4)
Smoker	1 (5.6)	0 (0.0)	1 (3.2)	0 (0.0)
AREDS formulation supplementation, n (%)				
No	17 (94.4)	12 (92.3)	6 (19.4)	2 (22.2)
Yes	1 (5.6)	1 (7.7)	25 (80.6)	7 (77.8)
Reticular Pseudodrusen, n (%)				
No	16 (88.9)	8 (61.5)	7 (22.6)	5 (55.6)
Yes	2 (11.1)	5 (38.5)	24 (77.4)	4 (44.4)
RIT, mean ± SD	5.1 (2.8)	6.7 (5.0)	15.6 (5.3)	12.1 (6.8)
AUDAC, mean ± SD	0.06 (0.04)	0.08 (0.07)	0.19 (0.11)	0.21 (0.19)

AREDS = Age-Related Eye Disease Study; AUDAC = area under the dark adaptation curve BMI = body mass index; OD = right eye; OS = left eye; RIT = rod-intercept time.

**Table 2 metabolites-11-00183-t002:** Metabolites associated with RIT (*p*-value < 0.01).

Super Pathway	Sub Pathway	Metabolite	Coefficient	*p*-Value
Amino Acid	Glutamate Metabolism	N-acetylglutamine	17.41	0.005
Amino Acid	Leucine, Isoleucine and Valine Metabolism	N-acetylleucine	21.34	0.008
Carbohydrate	Fructose, Mannose and Galactose Metabolism	mannitol/sorbitol	8.14	0.003
Lipid	Fatty Acid Metabolism (Acyl Choline)	linoleoylcholine	−16.55	0.008
Lipid	Medium Chain Fatty Acid	10-undecenoate (11:1n1)	−17.21	0.006
Lipid	Medium Chain Fatty Acid	5-dodecenoate (12:1n7)	−14.50	0.005
Lipid	Polyunsaturated Fatty Acid (n3 and n6)	linoleate (18:2n6)	−23.98	0.004
Lipid	Polyunsaturated Fatty Acid (n3 and n6)	linolenate [alpha or gamma; (18:3n3 or 6)]	−17.89	0.001 *

* Significant at the ENT80 threshold (*p*-value < 0.0019).

**Table 3 metabolites-11-00183-t003:** Metabolites associated with AUDAC (*p*-value < 0.01).

Super Pathway	Sub Pathway	Metabolite	Coefficient	*p*-Value
Amino Acid	Glutamate Metabolism	N-acetylglutamine	0.35	0.005
Amino Acid	Leucine, Isoleucine and Valine Metabolism	N-acetylleucine	0.46	0.004
Carbohydrate	Fructose, Mannose and Galactose Metabolism	mannitol/sorbitol	0.18	6.5 × 10^−^^4^ *
Lipid	Fatty Acid Metabolism (Acyl Choline)	linoleoylcholine	−0.39	0.002 *
Lipid	Fatty Acid Metabolism (Acyl Choline)	stearoylcholine	−0.34	0.005
Lipid	Hexosylceramides (HCER)	glycosyl ceramide (d18:2/24:1, d18:1/24:2)	−0.55	0.002 *
Lipid	Lysophospholipid	1-stearoyl-GPC (18:0)	−0.88	0.002
Lipid	Phosphatidylcholine (PC)	1-linoleoyl-2-linolenoyl-GPC (18:2/18:3)	−0.30	3.6 × 10^−^^4^
Lipid	Sphingomyelins	palmitoyl sphingomyelin (d18:1/16:0)	−1.26	0.008
Lipid	Sphingomyelins	sphingomyelin (d18:1/22:2, d18:2/22:1, d16:1/24:2)	−0.52	0.003
Lipid	Sphingomyelins	sphingomyelin (d18:1/24:1, d18:2/24:0)	−0.72	0.01
Lipid	Sphingomyelins	sphingomyelin (d18:2/23:1)	−0.46	0.003
Nucleotide	Purine Metabolism, (Hypo)Xanthine/Inosine containing	xanthine	−0.30	0.008
Nucleotide	Pyrimidine Metabolism, Uracil containing	beta-alanine	0.78	0.002 *

* Significant at the ENT80 threshold (*p*-value < 0.0019).

**Table 4 metabolites-11-00183-t004:** Metabolites associated with AUDAC for patients with RIT > 20 min (*p*-value < 0.01).

Super Pathway	Sub Pathway	Metabolite	Coefficient	*p*-Value
Amino Acid	Tryptophan Metabolism	indole-3-carboxylic acid	−0.50	0.007
Amino Acid	Tryptophan Metabolism	kynurenate	0.64	0.008
Lipid	Hexosylceramides (HCER)	glycosyl-N-palmitoyl-sphingosine (d18:1/16:0)	−0.87	0.007

**Table 5 metabolites-11-00183-t005:** Metabolites associated with RIT for patients with intermediate AMD (*p*-value < 0.01).

Super Pathway	Sub Pathway	Metabolite	Coefficient	*p*-Value
Amino Acid	Leucine, Isoleucine and Valine Metabolism	3-methyl-2-oxovalerate	47.36	0.001 *
Amino Acid	Leucine, Isoleucine and Valine Metabolism	3-methylglutaconate	19.97	4.9 × 10^−4^ *
Amino Acid	Leucine, Isoleucine and Valine Metabolism	3-methylglutarylcarnitine	12.37	0.007
Amino Acid	Leucine, Isoleucine and Valine Metabolism	isoleucine	69.79	5.0 × 10^−4^ *
Amino Acid	Leucine, Isoleucine and Valine Metabolism	leucine	73.71	0.002
Carbohydrate	Fructose, Mannose and Galactose Metabolism	mannitol/sorbitol	7.50	0.009
Lipid	Fatty Acid, Dicarboxylate	octadecanedioate (C18-DC)	−32.68	0.005
Lipid	Lactosylceramides (LCER)	lactosyl-N-nervonoyl-sphingosine (d18:1/24:1)	−53.85	0.007
Lipid	Polyunsaturated Fatty Acid (n3 and n6)	linolenate [alpha or gamma; (18:3n3 or 6)]	−27.20	0.008
Nucleotide	Purine Metabolism, (Hypo)Xanthine/Inosine containing	urate	52.09	0.006
Peptide	Gamma-glutamyl Amino Acid	gamma-glutamylisoleucine	25.66	0.01

* Significant at the ENT80 threshold (*p*-value < 0.0019).

## Data Availability

All tables and figures are original and have not been taken from any publication.

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
