# Peer review of "Association of Human Plasma Metabolomics with Delayed Dark Adaptation in Age-Related Macular Degeneration"

_metabolites, 2021, doi:10.3390/metabo11030183_

Round 1

Reviewer 1 Report

The manuscript entitled " Association of Human Plasma Metabolomics with Delayed Dark Adaptation in Age-Related Macular Degeneration"  presents results on the analysis of plasma samples in patients with macular degeneration. 

In general, the manuscript is easy to understand and follows a logical sequence. The introduction is clear and the methodology (materials and methods) provides enough information on the experimental work. There is a good level of detail in the description of the methods and the validation of the method.

Despite this, I would like to address some minor comments:

  • Regarding the significance criteria. Sometimes it’s defined at a p-value 0.01 and in other occasions, the significance is fixed at other p-values. I would encourage the authors to clary better this point at the materials and methods section.

In Tables 2, 3, 4 and 5 there is a column with p-values < 0.01 and then the authors use a code with *, **, and *** using a different threshold of significance. Please clary this point, which make the understanding of the tables confusing.

Some of the compound names in the column called “Metabolite” in Table 2 and 3 contain as well an *, which does not match with the previously mentioned criteria of significance, making the above mention even more confusing.

  • At the discussion, lines 187-190, it would be good to discuss and clary the relevance of this finding.
  • Please revise all the abbreviations used. Make sure to define them first time are used, and only need to be defined once (i.e. AUDAC and RIT are defined at the introduction and at the materials and methods section; AREDS is defined first time at materials and methods but used before in previous sections).
  • Please check and modify if appropriate the Institutional Review Board Statement section.
  • Please check and modify if appropriate the Informed Consent Statement section.

In summary, this is a well-written paper and the experiments were carefully planned. Some minor aspects should be clarified to improve the manuscript. After minor revisions, this paper could be considered for publication in Metabolites.

Author Response

I would like to thank you for the detailed review of our manuscript and very helpful suggestions. We revised our manuscript to address all the issues raised. Our point-by-point responses are in red and are as follows:

The manuscript entitled " Association of Human Plasma Metabolomics with Delayed Dark Adaptation in Age-Related Macular Degeneration" presents results on the analysis of plasma samples in patients with macular degeneration. 

In general, the manuscript is easy to understand and follows a logical sequence. The introduction is clear and the methodology (materials and methods) provides enough information on the experimental work. There is a good level of detail in the description of the methods and the validation of the method.

Despite this, I would like to address some minor comments:

  • Regarding the significance criteria. Sometimes it’s defined at a p-value 0.01 and in other occasions, the significance is fixed at other p-values. I would encourage the authors to clary better this point at the materials and methods section.

In Tables 2, 3, 4 and 5 there is a column with p-values < 0.01 and then the authors use a code with *, **, and *** using a different threshold of significance. Please clary this point, which make the understanding of the tables confusing.

Thank you for this comment regarding the significance criteria and the lack of clarity. To this end, we have removed the multiple thresholds of significance from the paper and kept two. A p-value < 0.01 to denote a trend towards significance, and a p-value < 0.0019 to denote statistically significant results after accounting for multiple corrections. A p-value < 0.01 was used due to the relatively low sample size, but as this increases the risk of false positive results, we also wanted to provide the reader an alternative p-value that denotes statistical significance.

To summarize, we:

  • Removed the multiple statistical significance thresholds, and kept p-value < 0.01 and p-value < 0.0019
  • Added an additional paragraph to explain both thresholds (line 78 – 82)
  • Removed the alternative significance thresholds from table 2-5
  • Updated the results (line 83-112)
  • Updated the materials and methods (line 583 – 587)

Some of the compound names in the column called “Metabolite” in Table 2 and 3 contain as well an *, which does not match with the previously mentioned criteria of significance, making the above mention even more confusing.

Thank you for pointing this out. Those * in the compound name added to the confusion, should not be there, and have now been removed.

  • At the discussion, lines 187-190, it would be good to discuss and clary the relevance of this finding.
    • We have included the line to clarify this finding, “suggesting these pathways plays an important role driving dark adaptation impairment at this stage of disease” (line 280-281)
  • Please revise all the abbreviations used. Make sure to define them first time are used, and only need to be defined once (i.e. AUDAC and RIT are defined at the introduction and at the materials and methods section; AREDS is defined first time at materials and methods but used before in previous sections).
    • Thank you for catching this. We have now revised all the abbreviations.
  • Please check and modify if appropriate the Institutional Review Board Statement section.
    • We have updated the Institutional Review Board Statement.
  • Please check and modify if appropriate the Informed Consent Statement section.
    • We have updated the Informed Consent Statement.

In summary, this is a well-written paper and the experiments were carefully planned. Some minor aspects should be clarified to improve the manuscript. After minor revisions, this paper could be considered for publication in Metabolites.

Thanking you,

Sincerely,

Deeba Husain MD

Reviewer 2 Report

Dear Editor

Thank you for the opportunity to review the interesting manuscript. Mendez et al. presented their original work reporting the association between plasma metabolomics and delayed dark adaptation in age-related macular degeneration. This work has originality and novelty. Several concerns should be addressed before publication.

Previous studies reported that DA impairment is associated with AMD severity and reticular pseudodrusen (RPD). (Ref.62 Flamendorf et al 2015, Ophthalmology) How many patients with RPD are there in each group?

Subdivide into RPD (+) or RPD (-) and analyze the association between RPD and the associated metabolomics.

Nine patients with late AMD were included in this study. How many patients with central GA or neovascular AMD were there? In addition, did patients with nAMD have active lesions or scarring?

Author Response

I would like to thank you for the detailed review of our manuscript and very helpful suggestions. We revised our manuscript to address all the issues raised. Our point-by-point responses are in red and are as follows:

Thank you for the opportunity to review the interesting manuscript. Mendez et al. presented their original work reporting the association between plasma metabolomics and delayed dark adaptation in age-related macular degeneration. This work has originality and novelty. Several concerns should be addressed before publication.

Previous studies reported that DA impairment is associated with AMD severity and reticular pseudodrusen (RPD). (Ref.62 Flamendorf et al 2015, Ophthalmology) How many patients with RPD are there in each group?

Thank you for this comment on reticular pseudodrusen (RPD). Due to the importance of RPD, we have included it in Table 1 - Clinical and demographic characteristics of the study cohort.

Subdivide into RPD (+) or RPD (-) and analyze the association between RPD and the associated metabolomics.

Thank you for this insight into an important stratified analysis in this paper. Unfortunately, with the imbalance of RPD (+/-) by AMD Stage (in particular – intermediate AMD) and the already small sample sizes, we were not able to perform a suitable stratified analysis in this study. We have included this in our limitation in the discussion, line 406-411.

Nine patients with late AMD were included in this study. How many patients with central GA or neovascular AMD were there? In addition, did patients with nAMD have active lesions or scarring?

Thank you for this comment on late AMD patient. We have included a sentence at line 75-76, “Of the 9 late AMD patients, 5 had geographic atrophy (GA) and 4 had choroidal neovascularization (CNV) with active lesions.”

Sincerely,

Deeba Husain MD

Reviewer 3 Report

This study by Mendez et al. analyzed the serum metabolites of different groups of patients suffering from AMD. The most significant aspect of this work is the fact that they are stratifying patients based on the earliest functional problem in AMD, dark adaptation. I congratulate the authors to do this. Omics studies are often suffering from the same problem: insufficient patient numbers. This study is not different. The cost of doing metabolomics makes it impossible to make it large enough unless restrictions are introduced, and I do not intend to mention this as a fact that should stop publication. I strongly support studies even with low numbers to be published so others can learn from them. The findings in this study are very interesting and if there is one problem then the omission of referencing a body of evidence especially generated by the Eye-Risk Consortium in recent years. I copied below references that could, and probably should, b included as discussion in the relationship of these findings are highly relevant for this publication:

https://www.aaojournal.org/article/S0161-6420(20)30561-3/fulltext

https://pubmed.ncbi.nlm.nih.gov/30315903/

https://pubmed.ncbi.nlm.nih.gov/30591665/

https://pubmed.ncbi.nlm.nih.gov/30114418/

https://pubmed.ncbi.nlm.nih.gov/31220133/

https://www.tandfonline.com/doi/full/10.1080/23808993.2018.1502037

This is not an exhaustive list, just some that I know of. I do believe it is critical to refer to these or some of these here.

One of the issues I would like to ask the Authors to change is the reference to "AREDS usage". This is completely inappropriate as AREDS is a reference to a huge study and not to a product. There is the AREDS formulation, although patients are unlikely to walk into a shop and buy AREDS formula.....

Another point I would like to enquire about is the potential issue of correcting the models for the AREDS formula intake. I would expect that supplementation would alter the outcome of the metabolomics measures, and I am not convinced that adjusting the analysis actually necessarily valid, though I am not an expert. I personally would be really interested in seeing the differences in metabolites between those who are taking the supplement and those who do not. I know that numbers are small, but this would be very important.

Please rewrite the section from line 340- 351, as someone who does this for a living I could not understand what the Authors did. The circle of 433 mm in an eye does not seem appropriate.

Clarify how individual eyes were handled statistically? One patient has one condition and one blood sample and it is not clear to me whether this was appropriately addressed. While I am not a statistics expert This has to be fully proofed before the conclusion.

Author Response

I would like to thank you for the detailed review of our manuscript and very helpful suggestions. We revised our manuscript to address all the issues raised. Our point-by-point responses are in red and are as follows:

This study by Mendez et al. analyzed the serum metabolites of different groups of patients suffering from AMD. The most significant aspect of this work is the fact that they are stratifying patients based on the earliest functional problem in AMD, dark adaptation. I congratulate the authors to do this. Omics studies are often suffering from the same problem: insufficient patient numbers. This study is not different. The cost of doing metabolomics makes it impossible to make it large enough unless restrictions are introduced, and I do not intend to mention this as a fact that should stop publication. I strongly support studies even with low numbers to be published so others can learn from them. The findings in this study are very interesting and if there is one problem then the omission of referencing a body of evidence especially generated by the Eye-Risk Consortium in recent years. I copied below references that could, and probably should, b included as discussion in the relationship of these findings are highly relevant for this publication:

https://www.aaojournal.org/article/S0161-6420(20)30561-3/fulltext

https://pubmed.ncbi.nlm.nih.gov/30315903/

https://pubmed.ncbi.nlm.nih.gov/30591665/

https://pubmed.ncbi.nlm.nih.gov/30114418/

https://pubmed.ncbi.nlm.nih.gov/31220133/

https://www.tandfonline.com/doi/full/10.1080/23808993.2018.1502037

This is not an exhaustive list, just some that I know of. I do believe it is critical to refer to these or some of these here.

Thank you for spotting this. It was not our intent to omit the large body of evidence generated by the Eye-Risk Consortium. We have updated to discussion to reflect this: (1) adding additional sentences in the discussion (lines 319-320, 341-343), and (2) adding the references where they should have been included (lines 302-303, 304).

One of the issues I would like to ask the Authors to change is the reference to "AREDS usage". This is completely inappropriate as AREDS is a reference to a huge study and not to a product. There is the AREDS formulation, although patients are unlikely to walk into a shop and buy AREDS formula.....

Thank you for the comment, we realize that this wording was inappropriate. We have changed “AREDS usage” to “AREDS formulation supplementation”.

Another point I would like to enquire about is the potential issue of correcting the models for the AREDS formula intake. I would expect that supplementation would alter the outcome of the metabolomics measures, and I am not convinced that adjusting the analysis actually necessarily valid, though I am not an expert. I personally would be really interested in seeing the differences in metabolites between those who are taking the supplement and those who do not. I know that numbers are small, but this would be very important.

This is a great comment, and something we should have previously included in the discussion regarding limitations (now at line 399-406). While we also wanted to look at a stratified analysis based on supplementation usage, there is an imbalance of AREDs formula supplement use by AMD stage (low supplementation in controls and early AMD, and high supplementation in intermediate and late AMD). Unfortunately, with this imbalance, we need a larger sample size to perform a suitable stratified analysis.

Nutrition, in this case, vitamins (via supplementation) and it’s derivates, can be directly measured in the blood with metabolomic platforms and importantly affect multiple interconnected metabolomic pathways. Without correcting for AREDS formula supplementation, due to the imbalance of usage by AMD stage, i.e. high supplementation in intermediate and late AMD, we could find erroneous results.

In a targeted analysis, analyzing the association between plasma vitamin E metabolites and RIT adjusting for age, smoking status, race, BMI and AMD stage (and not AREDs formulation supplementation), we find gamma-tocopherol/beta-tocopherol (p-value = 0.01) and alpha-tocopherol (p-value = 0.03) significant.

Please rewrite the section from line 340- 351, as someone who does this for a living I could not understand what the Authors did. The circle of 433 mm in an eye does not seem appropriate.

Thank you for catching this. This was supposed to be 433 μm, not 433 mm. We edited the section on AMD grading to make it more clear as suggested by the reviewer.

Clarify how individual eyes were handled statistically? One patient has one condition and one blood sample and it is not clear to me whether this was appropriately addressed. While I am not a statistics expert This has to be fully proofed before the conclusion.

Thank you for pointing this out, we updated the sentence at line 571-572 to make this clear, i.e., using a multi-level linear mixed model to account for two dark adaptation measures with a single plasma metabolite level for the patient. This model approach has been previously used in our group to look at structural changes and peripheral changes with dark adaptation (reference 2, 4). Additionally, we used a similar approach (i.e. one blood sample and AMD stage per eye) in a previous metabolomics-based study (reference 9).

Sincerely,

Deeba Husain MD